

# On the assessment of the moisture transport by the Great Plains low-level jet

Iago Algarra[1], Jorge Eiras-Barca[1], Gonzalo Miguez-Macho[2], Raquel Nieto[1], and Luis Gimeno[1]

[1]EPhysLab (Environmental Physics Laboratory), Facultade de Ciencias, Universidade de Vigo, Ourense, Galicia, Spain
[2]Non-Linear Physics Group, University of Santiago de Compostela, Galicia, Spain

*Correspondence to*: Iago Algarra (ialgarra@uvigo.es)

**Abstract.** Low-Level Jets (LLJs) can be defined as filamentous wind corridors of anomalously high wind speed values located within the first km of the troposphere. These structures, together with atmospheric rivers (ARs), are the major meteorological systems in the meridional transport of moisture on a global scale. In this work, we focus on the Great Plains low-level jet, which plays an important role in the moisture transport balance over the central United States. The Gulf of Mexico is the main moisture source for the GPLLJ, which has been identified as a key factor for rainfall modulation over the eastern and central US.

The relationship between moisture transport from the Gulf of Mexico to the Great Plains and precipitation is well documented in previous studies. Nevertheless, a large uncertainty still remains in the quantification of the moisture amount actually carried by the GPLLJ. The main goal of this work is to address this question. For this purpose, a relatively new tool, the regional atmospheric Weather Research and Forecasting Model with 3D water vapour tracers (WRF-TT, Insua-Costa and Miguez-Macho, 2018) is used together with the Lagrangian model FLEXPART to estimate the load of precipitable water advected within the GPLLJ. From a climatology of jet intensity over a 37-year period (Rife et al., 2010), which follows a Gaussian distribution, we select for study 5 cases representing the mean, and one and two standard deviations above and below it. Results show that the jet is responsible for roughly 70%-80% of the moisture transport occurring in the southern Great Plains when a jet event occurs. Furthermore, moisture transport by the GPLLJ extends to the northeast US, accounting for 50% of the total in areas near the Great Lakes. Vertical distributions show the maximum of moisture advected by the GPLLJ at surface levels and maximum values of moisture flux about 500 m above, in coincidence with the wind speed profile.

## 1 Introduction

It is well known that the Great Plains Low-Level Jet (hereafter, GPLLJ) plays an important role in the balance of the moisture transport over the central United States (Schubert et al., 1998; Stensrud, 1996). The atmospheric moisture is transported by the GPLLJ from tropical and subtropical latitudes (particularly the Gulf of Mexico and the Caribbean Sea) into the Great Plains (Helfand and Schubert, 1995; Mo et al., 1997) where the jet is often responsible for nocturnal deep convective activity (Higgins



et al., 1997a; Pu et al., 2016). In the last decades, a large number of authors have shown the strong influence of the GPLLJ as a modulator of climate and rainfall over this region and even further east (Byerle and Paegle, 2003; Mo et al., 1995b, 1997; Wu and Raman, 1998); for instance, throughout May and June it is estimated that at least one-third of the moisture penetrating into the continental US is carried by the GPLLJ (Helfand and Schubert, 1995).

Among the mechanisms which have been proposed as triggers of the GPLJJ are included a combination of inertial oscillations (Blackadar, 1957) and orographic forcing (Byerle and Paegle, 2003; Pan et al., 2004; Ting and Wang, 2006; Wexler, 1961). Particularly, the mechanism of Blackadar (1957) suggests that inertial oscillations near the friction layer can induce the formation of the GPLLJ (Wu and Raman, 1998). Nevertheless, orographic effects are also understood as a key factor in the maintenance of the GPLLJ. In this regard, Ting and Wang (2006) proved that, when the interaction with the orography is

removed from numerical simulations, the GPLLJ vanishes, together with an important amount of the summer precipitation over the central and southern US.

The GPLLJ is a phenomenon confined within the first kilometres of the troposphere and is closely related to the warm season (Bonner, 1968). Besides, it is characterized by a strong diurnal oscillation, with a peak in strength during night hours (Augustine and Caracena, 1994). The GPLLJ is a phenomenon extremely localized in time and space and its role in the

continental moisture balance is difficult to study solely from observations.

Nevertheless, a large number of studies have documented the relationship between the major moisture transport and the GPLLJ. Higgins et al., (1996) studied the moisture budget over the central US in May employing NASA/DAO and NCEP/NCAR datasets, together with station observations, to evaluate the limitations of these products. Although both reanalyses overestimate daily mean precipitation rates, they accurately capture the basic temporal and structural characteristics of the

GPLLJ. From the data, these authors calculated an increase in atmospheric moisture transport from the Gulf of Mexico during nightime of more than 50%. In a later work, Higgins et al., (1997b) observed a well-defined nocturnal maximum of precipitation over the Great Plains in spring and summer by analysing station data. Particularly, this research found over the region an excess of 25% in nocturnal precipitation during summer when compared with the diurnal one, associated with a rainfall decrease over the Gulf of Mexico. Additionally, this work reveals significant differences in precipitation pattern in

coincidence (or not) with LLJ events. When a LLJ event occurs, the observations show an enhanced precipitation over the north-central United States and the Great Plains region, together with a decrease along the Gulf of Mexico and the western Atlantic (Mo et al., 1997). On the other hand, Mo and Juang (2003) found a regional dependence between evaporation and precipitation, reflected in evaporation anomalies over the Great Plains along the trajectory of the GPLLJ, which are associated with downstream precipitation anomalies.

All these results are consistent with the large-scale atmospheric moisture transport and support the marked influence of the GPLLJ over the central-eastern US, which has been shown to trigger more than 60% of the spring local precipitation there (Wang and Chen, 2009).

Otherwise, extreme rainfall events in the central US are related to an increase in moisture convergence downwind of the GPLLJ (Mo et al., 1997). Thus, important socioeconomic impacts follow enhanced GPLLJ events, which modulates a large percentage



of the local extreme precipitation events and flooding in warmer months (Arritt et al., 1997; Beljaars et al., 1996; Mo et al., 1995a, 1997; Trenberth and Guillemot, 1996).

During the last decades, the GPLLJ has experienced a strengthening, accompanied by a northward migration causing a displacement of rainfall in the same direction. As a result, more common droughts have been observed in the southern Great Plains (Barandiaran et al., 2013).

The increase in the number and intensity of GPLLJ events is also forecasted for future projections, which reveal an intensification of the GPLLJ during the spring season associated with global warming (Cook et al., 2008). As a result, increasing amounts of moisture transport and rainfall are expected, particularly from April to July, over the central US (Harding and Snyder, 2014). The same projections forecast a slight weakening of the GPLLJ from August to December, which could translate into increasing drought conditions.

The knowledge about the GPLLJ, together with the insights on the relationship between the moisture transported by the GPLLJ and local precipitation patterns has increased considerably during the last decades. However, there are still unanswered questions about the quantification of such water vapour transport and specially about the estimation of the ratio of land to oceanic moisture sources associated with the GPLLJ. This estimate of the oceanic input to the moisture transport associated with the GPLLJ is essential to predict and understand the behaviour of the GPLLJ in future scenarios.

In this work, a new tool, the regional atmospheric Weather Research and Forecasting Model with 3D water vapour tracer diagnostics (WRF-TT, Eiras-Barca et al., 2017; Insua-Costa and Miguez-Macho, 2018) is used to quantify the total amount of total precipitable water (TPW) transported by the GPLLJ. To show the differences between the transport of moisture on jet and non-jet days, a 37-year climatology was calculated previously and the at the point of maximum jet intensity is obtained following the methodology by Rife et al., (2010). The structure of this work is as follows, in Section 2 we provide the methodology used, in Section 3 we show the results obtained, and finally in Section 4 we discuss conclusions.

## 2 Data and methods

### 2.1 Detection of the Great Plains low-level jet

To objectively detect days with LLJ over the Great Plains, we applied the night-time index proposed in Rife et al., (2010), hereafter named as NLLJ. This index is based on the temporal variation of the wind's vertical structure and the fact that LLJs are most intense at local midnight. Because both frequency and intensity of GPLLJ are mostly associated with the warm season, we develop a 37-year climatology for the month of July (boreal summer). According to the NLLJs characteristics, and with the aim to define the index, two conditions should be met to consider a GPLLJ detection:

1. The wind speed is higher at local midnight than at midday.
2. The local midnight wind speed is higher at the surface (~ 500m) than in height (~ 4km).

The index is calculated at each grid point over an area centred over the US using 6-hourly ERA-Interim reanalysis data (Dee et al., 2011) with a 0.25º horizontal resolution. Due to the jet core is located within of the first kilometre of the troposphere, it



is necessary to take into account the elevation of the land, so sigma coordinates are used. The GPLLJ-climatology is developed for 37 years, from 1980 to 2016, and the NLLJ index can be defined as follows:

$$NLLJ = \lambda\varphi\sqrt{[(u_{00}^{L1} - u_{00}^{L2}) - (u_{12}^{L1} - u_{12}^{L2})]^2 + [(v_{00}^{L1} - v_{00}^{L2}) - (v_{12}^{L1} - v_{12}^{L2})]^2} \quad (1)$$

where $u$ and $v$ are the zonal and meridional wind components, respectively. *L1* represents the winds at the surface at 53 sigma level (elevation near the jet core), approximately 500 m above ground level (AGL), while *L2* corresponds to the wind at 42 sigma level (around 4000 m AGL). Numbers *00* and *12* refer to local midnight and local noon, respectively. $\lambda$ and $\varphi$ are binary multipliers representing the temporal and vertical variation of 5 the wind. Particularly, $\lambda$ relates to the daily variation of the

10 wind at 500 m and $\varphi$ refers to the wind's vertical variation between 500 m and 4 km at midnight (Rife et al., 2010):

$$\lambda = \begin{cases} 0, ws_{00}^{L1} \le ws_{12}^{L1} \\ 1, ws_{00}^{L1} > ws_{12}^{L1} \end{cases} \quad (2)$$

$$\varphi = \begin{cases} 0, ws_{00}^{L1} \le ws_{00}^{L2} \\ 1, ws_{00}^{L1} > ws_{00}^{L2} \end{cases} \quad (3)$$

**2.2 Identification of moisture sources associated with Great Plains low-level jet**

For the objective identification of moisture sources associated with the GPLLJ, the Lagrangian backward trajectories from the FLEXPART v9.0 model are used (Stohl et al., 2004; Stohl and James, 2005). This model provides a large number of air parcel trajectories from which it is possible the calculation of the evaporation minus precipitation budget, tracking all changes in the specific humidity of air parcels.

FLEXPART has been widely and successfully used to track moisture paths for the study of the atmospheric branch of the

20 hydrologic cycle in different parts of the world (e.g., Hu et al., 2018; Sorí et al., 2018; Vázquez et al., 2016). Furthermore, this tool is able to infer the moisture sources for precipitation falling in a target region when backward trajectories are considered (eg., Drumond et al., 2010; Durán-Quesada, 2012; Gimeno et al., 2012; Ramos et al., 2016; Stohl et al., 2008; Wegmann et al., 2015).

In this work we use the outputs of a global experiment in which FLEXPART v9.0 tracks approximately 2 million particles (air

parcels) with constant mass distributed on the globe every time step during 37-year period (1980-2016). These air parcels are advected by the 3D wind field, and the variables of interest of each particle (such as position, height, specific humidity, temperature among many others) are saved at each time step. We perform a FLEXPART simulation fed with ERA-Interim reanalysis data at 1º horizontal resolution on 61 vertical levels from sea level pressure to 0.1 hPa and 6-hour time intervals (00, 06, 12 and 18 UTC). The model is run with a 3 h timestep, and linear interpolation is used to obtain data with this frequency

from ERA-Interim. The backward trajectories are followed during 10 days, which is the average life time of water vapor in the atmosphere (Numaguti, 1999).



The changes in specific humidity (q) of each air parcel along its path can be expressed as follows:

$$e - p = m \frac{dp}{dt} \quad (4)$$

where $m$ is the mass of a particle (which remains constant in the simulation), $q$ is the specific humidity, $t$ the time step, and $e$ - $p$ (evaporation minus precipitation) represents the water flux associated with the particle. To obtain the instantaneous values of the $E$ - $P$ balance in a given area (in this case, over one of 1.0 x 1.0 degrees in latitude and longitude), it is necessary to integrate the moisture changes for all particles present in the atmospheric column over such area ($E$ denotes evaporation and $P$ the precipitation rate per unit area). Thus, in a backward experiment, a moisture source is defined as those regions where $E$

- $P$ is positive $(E - P > 0)$, which implies that evaporation exceeds precipitation, while a moisture sink is defined as a region where $E$ - $P < 0$, meaning that precipitation is greater than evaporation.

In our study, backward trajectories were followed from the area composed of points with values of NLLJ above percentile 75.

## 2.3 The regional atmospheric Weather Research and Forecasting Model with 3D water vapour tracer diagnostics (WRF-TT)

The mesoscale Weather Research and Forecast model (WRF 3.8.1) with the moisture tracers tool (WRF-TT, Eiras-Barca et al., 2017; Insua-Costa and Miguez-Macho, 2018) is used to carry out to quantify the total amount of total precipitable water (TPW) transported by the GPLLJ. In order to analyze the moisture transport associated with the GPLLJ avoiding the effects of other synoptic-scale transport events, we tag the moisture passing northward through a narrow wall located on the northern edge of the moisture source region identified using the FLEXPART model. When a particle of water (whether in liquid, solid

or gas state) crosses the wall, it is labeled for further analysis inside the simulation domain. We consider all water traversing the wall to be advected by the GPLLJ.

The horizontal resolution of the simulations is 20 km and the vertical column is divided into 38 levels. The simulation covers a time window of 11 days, starting 7 days prior to the day of interest. The model parameterizations together with the WRF-TT are set using the PBL Yonssei University (YSU) parametrization (Hu et al., 2013; Shin and Hong, 2011), the schemes of Kain-

Fritsch for convection (Kain, 2004), the Dudhia one for short-wave radiation (Dudhia, 1989), the Rapid Radiative Transfer Model (RRTM) svjeme for long-wave radiation (Mlawer et al., 1997), and the WRF Single-Moment 6-Class Microphysics Scheme (WSM6) (Hong and Lim, 2006).

In addition, we apply spectral nudging of waves longer than 1000 km above the boundary layer, with a relaxation time of 1h, to avoid 30 distortion of the large-scale circulation. This configuration has been validated and successfully applied several

30    times with the WRF-TT in mid latitudes (e.g., Eiras-Barca et al., 2017). Spectral nudging ensures that the large-scale circulation is well captured in the simulations. ERA-Interim data provides lateral boundary and initial conditions for the runs (Dee et al., 2011). The variables of interest for the analysis of the GPLLJ event are computed as follows. Integrated Water Vapor (IWV), Eq. (5) is obtained by vertical integration of the specific humidity (q) in pressure (p) levels, where g represents the gravitational



force. The instant flux of moisture ($\varphi$) is calculated as stated in Eq. (6) and the conversion between (g) and the water vapor mixing ratio obtained from WRF is performed using Eq. (7), where u and v are the horizontal components of the wind field.

$$IWV_{(i,j)} = \frac{1}{g(k)} \int_{surface}^{top} q(i,j,k)dp \qquad (5)$$

$$\varphi(i,j,k) = |q \cdot (u,v)| \qquad (6)$$

$$q = \frac{w}{w+1}, with\ w \ll 1 \rightarrow q \approx w \qquad (7)$$

## 3 Results

### 3.1 Characterization of the Great Plains low-level jet

As previously mentioned, the NLLJ index was calculated at each grid point over the North American region for the month of July over the period 1980-2016. Fig. 1.a shows the climatological NLLJ index and the wind field at 500 hPa. The black cross indicates the point of maximum intensity of the index (8.8 m s$^{-1}$). At this point, located at 32.75ºN-99ºW, along the 37-years analysed, and for July, a total of 931 LLJ days are identified, that is, 81% of all days have a positive value of the index. On the point of maximum intensity showed in Fig1.a, Fig1.b displays the frequency distribution of the NLLJ for the period 1980-2016. A clear peak around 11 m s$^{-1}$ is observed together with a Gaussian behaviour (R$^2$ = 0.95, red line). The latter has been used to select the five case studies to be analized with WRF-TT and listed in Table 1. The five events chosen correspond to $\mu$, $\mu\pm2\sigma$, $\mu\pm\sigma$ (where $\mu$ is the mean of the distribution and $\sigma$ its standard deviation), and they provide a general perspective of the LLJ's behaviour. Since each case-study WRF-TT simulation spans for 11 days, a condition of persistence of the index value for at least two days after the main jet day is applied. Additionally, we perform a sixth simulation with a non-jet day (simulation 0 in the Table 1). This non-jet day is selected from the developed climatology as the longest non-jet period, in order to avoid overlaps in moisture transport with jet days.

### 3.2 Moisture transport associated with the Great Plains low-level jet

In order to detect the main oceanic source of moisture for the GPLLJ we used the FLEXPART trajectories outputs. The area englobed in the 75$^{th}$ percentile of the LLJ index values (cyan line in figure 1.a) was selected as the target region for the backward experiment (as it was explained in the methodology). Fig. 2 shows the source of moisture in red color, obtained as the 75$^{th}$ percentile of the ($E - P > 0$) field. This area covers the southern Gulf of Mexico and extends into the Caribbean Sea, between 60º-98ºW and 12º-28ºN.

Although the flow originated in the source of moisture is advected in the low levels as a result of the strong intensity of the trade winds, a 3D-label wall (at 29ºN and from 94.5ºW to 100ºW) was used in the WRF-TT simulations (orange line in fig. 2). The position of the sentinel wall was selected on the region where oceanic moisture associated with the GPLLJ landfalls.

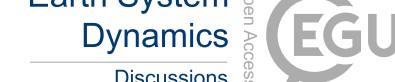



The wall remained constant in the WRF-TT simulations. A thin wall was used instead of the entire source regions in order to avoid overlaps in the labelling of moisture caused by secondary synoptic scale mechanisms.

Fig. 3 shows the ratio of precipitable water transported by the GPLLJ to total precipitable water *(TPW$_{tracers}$=TPW)* for the six case-studies analyzed. As mentioned earlier, *TPW$_{tracers}$* represents the TPW that has crossed the "wall" highlighted in orange

in Figure 1, which we assume corresponds to moisture advected by the GPLLJ. Following the same behaviour of the GPLLJ itself, the moisture flux is initially in the northward direction and veers east as it penetrates into the Great Plains for all events with positive NLLJ index values. As expected, the non-jet event with NLLJ value equal to zero (Sim 0) does not show remarkable moisture fluxes. For the jet events, ratios are close to one in regions near the tagging wall and extend for hundreds of km northward with significant values above 60%. Percentages between 70% and 80% are observed in the Great Plains. The

large geographical reach of the moisture associated with the GPLLJ is evidenced in this figure, showing that for certain GPLLJ events it can occasionally explain more than 50% of TPW even in the north-east US. It is necessary to highlight that higher values in the index value does not necessarily mean larger flows of moisture in the entire, as can be observed, for example, when SIM3 and SIM5 are compared.

As it was previously stated, the aim of this work is to study the general behaviour of the GPLLJ associated with its moisture

transport. In the first simulated case of GPLLJ (fig.3 – Sim 1) it is observed that most of the precipitable water is concentrated on the Great Plains, exceeding ratios of 80% out of the total**.** In the second GPLLJ event simulated (fig.3 – Sim 2), the precipitable water extends northeast of the US and to the south of the Great Lakes and the GPLLJ, where explain close of the 50% of precipitable water. The third simulated case corresponds to the average behaviour of the GPLLJ (fig.3 – Sim 3) and evidences the influence of the GPLLJ in the northeast of US with ratios near 50% in the US East Coast. Nevertheless, in areas

along the path of the GPLLJ, the advection of precipitable water is close to 80%. In the fourth and fifth simulations of GPLLJ (fig.3 – Sim 4 and 5), the plume of precipitable water is concentrated over the Great Plains. However, the water precipitable ratio is reduced as latitude increases, but values are still close to 50% in the norheastern areas of the US.

Fig. 4 shows the statistically weighted mean of the ratio shown in Fig. 3 for the five case studies with NLLJ > 0 considered in the analysis. The weights associated with each event are stated in Table 1 at the last column, and the objective criteria to assign

them can be found in appendix A1. The aim of using weights in the analysis is to give greater importance to the event representing the mean value of the NLLJ and less relevance to the events in the tail of the distribution. Notwithstanding the limited number of simulations used in the analysis, this procedure allows us to interpret Figure 4 as a "climatology" of the moisture transport associated with the GPLLJ. Roughly 80-90% of the precipitable water in its core zone of influence over the Great Plains, in Texas and Oklahoma, is carried by the GPLLJ when a jet event occurs. With increased distance from that area,

the ratio of precipitable water transported by the GPLLJ decreases; however, the contribution of moisture from the Gulf of Mexico to TPW is still of more than 50% as far north as the Great Lakes.

Fig. 5 shows the vertical distribution of tracer specific humidity (q$_{TR}$) and tracer water vapor flux (φ$_{TR}$) for cross sections at positions depicted in Fig. 6 for the main GPLLJ event (1992.07.11). Tracer moisture (a-c) has a maximum at surface levels, while the moisture flux (d- f) maximizes at 500 m AGL where the LLJ core is located. A significant presence of both tracer



water vapor and tracer water vapor flux is restricted to the first 2 km AGL. Overall, as the latitude increases the water vapor plume from the Gulf of Mexico tends to rise in the vertical column and expand zonally along the GPLLJ path to the east of the U.S. Equivalent conclusions can be obtained from the remaining events, which are shown in supplementary material S1.

## 4 Conclusions and discussions

A combination of Lagrangian and Eulerian methods were used to identify and objectively quantify the moisture transport associated with the Great Plains Low-Level Jet (GPLLJ). First, the langrangian model FLEXPART was used to locate the GPLLJ moisture sources in the Gulf of México for the month of its maximum activity (July) throughout the period 1980-2016. Once the Gulf of Mexico was identified as the main source of moisture ($E – P > 0$) for the GPLLJ, we use a new tool known as eulerian 3D WRF-WTT (Eiras-Barca et al., 2017; Insua-Costa and Miguez-Macho, 2018) which was applied to track the moisture advected in six selected GPLLJ events based on the distribution of the index used previously to detect the GPLLJ (Rife et al., 2010). This work analysed the behaviour of the GPLLJ during the month of its maximum activity (July) for a long period, 1980-2016, and we select six representative cases.

The moisture transport analysis reveals the major role played by the GPLLJ in the water cycle of central North America, transporting large amounts of moisture from the Gulf of Mexico as far as the north-east US. Particularly, advection by the jet explains more than 80% of the precipitable water in the southern Great Plains when a jet event occurs, which, in July, is most of the days. The Rocky Mountains blocks the circulation of GPLLJ and force it to turn to the east of the US, reaching even the eastern coast of the US. The moisture transport associated with the GPLLJ is in a plume of moisture where, the maximum transport occurs in the path of the GPLLJ. As expected, the ratio reduces as latitude increases, but values are still close to 50% in the norheastern areas of the US.

We note that the extension of the GPLLJ is dependent on the synoptic conditions, among other factors, which are out of the scope of this paper. For example, the presence of a well-developed high pressure system in higher latitudes of North America may block the advection of the GPLLJ moisture to this region. Dong et al., (2011) related the drought of 2006 with an anomalous high over south-western U.S region and an anomalous low over the Great Lakes. This pattern inhibited the advection of moisture from the Gulf of Mexico contributing to the extreme dryness, and the lack precipitation was associated with a suppressed cyclonic activity over the south-western US. However, the 2007 flood events were initially leaded by active synoptic weather patterns, linked to an active moisture transport from the Gulf of Mexico by the GPLLJ.

Besides, higher values in the NLLJ index mean larger differences between winds aloft and at the surface at the reference point, but do not necessarily mean stronger moisture transport.

Thus, results should be understood as a first approach to the quantification of the large extent of GPLLJ moisture advection and its implications for the water budget in North America. More WRF-TT simulations should be conducted, and other months should be included in FLEXPART backward calculations to extend this work and produce a more comprehensive analysis.



**Author contributions.**

RN, LG and IA had the initial idea. GMM developed WRF-WVT tool. RN, LG, IA and JEB carried out the simulations and data analysis. RN, LG and GMM provided suggestions, commented and reviewed the manuscript before submission.

**Competing interests.**

5    The authors declare that they have no conflict of interest.

**Acknowledgments.**

The ECMWF ERA-Interim reanalysis data were obtained from https://www.ecmwf.int/en/forecasts/datasets/ reanalysis-datasets/era-interim. Iago Algarra was financially supported by Spanish government (MINECO) (CGL2015-65141-R). Iago Algarra would like to express his gratitude to the all Non-Linear Physics Group for kind support during their stay in the

10    University of Santiago de Compostela.

25

30





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

25



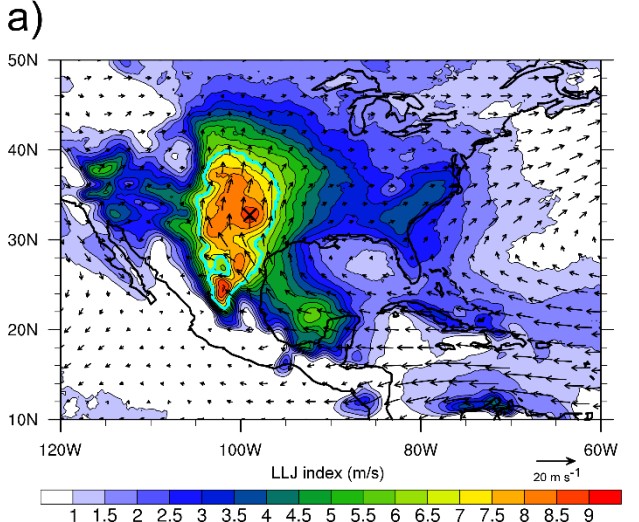

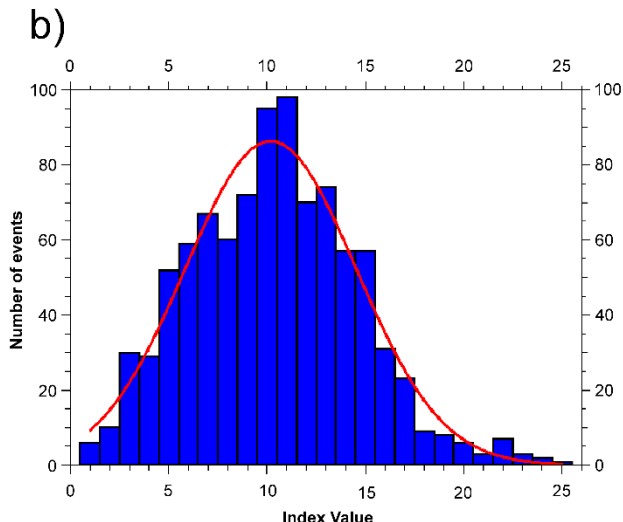

Figure 1: (a) Mean NLLJ index (shaded) and 500 m winds (arrows, in m s$^{-1}$) at local midnight in July (boreal summer) for 1980-2016, calculated from ERA-Interim reanalysis. The black cross at 32.75ºN, 99ºW shows the point of maximum NLLJ in the climatology. The cyan contour line surrounds the region containing points above the 75th percentile. (b) Frequency distributions of the GPLLJ for the months of July from 1980 to 2016 (blue bars). The red curve corresponds to the Gaussian fit: $y(x) = y_0 + A \cdot \exp\left(-\frac{(x-y)^2}{2\sigma^2}\right)$.



25

Figure 2: **Highlighted in red are moisture sources obtained with FLEXPART from backward trajectories originating in the region outlined in cyan in Fig 1.a. The orange line over the continent marks the position from where precipitable water is tagged in WRF-TT, corresponding to the northern edge of the FLEXPART source region. All water vapor and condensate crossing through that line is considered as moisture advected by the GPLLJ for further analysis.**





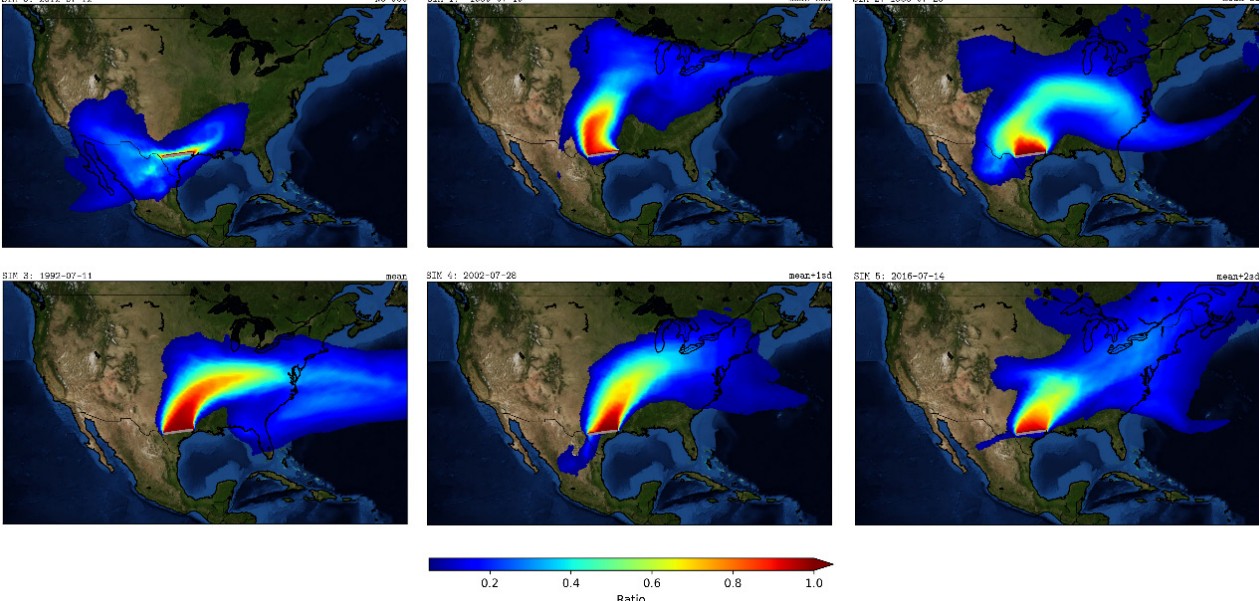

**Figure 3: Ratio of tagged precipitable water transported by the GPLLJ to total for the six case-studies analysed.**



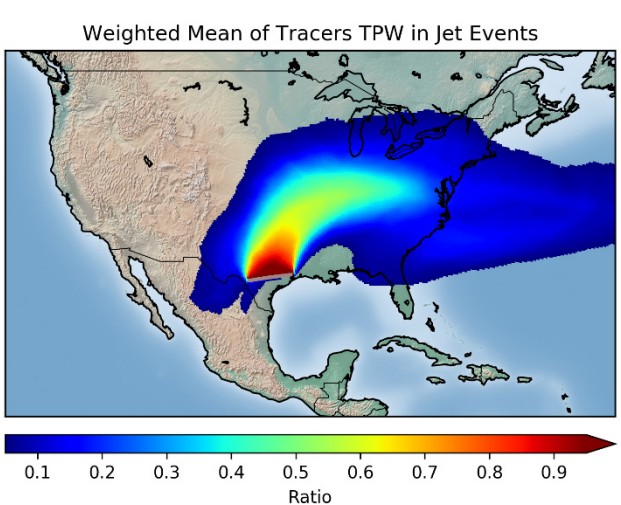

**Figure 4: Statistically weighted ratio of precipitable water transported by the GPLLJ for the five case studies with NLLJ > 0 considered in the analysis in Figure 3. Weights applied are stated in Table 1.**





**Figure 5: (a-c) qTR in g kg⁻¹ for the three vertical cross sections at the locations depicted with white lines in Fig. 6. (d-f) same as (a-c) but for φTR in g m (kg s)⁻¹.**





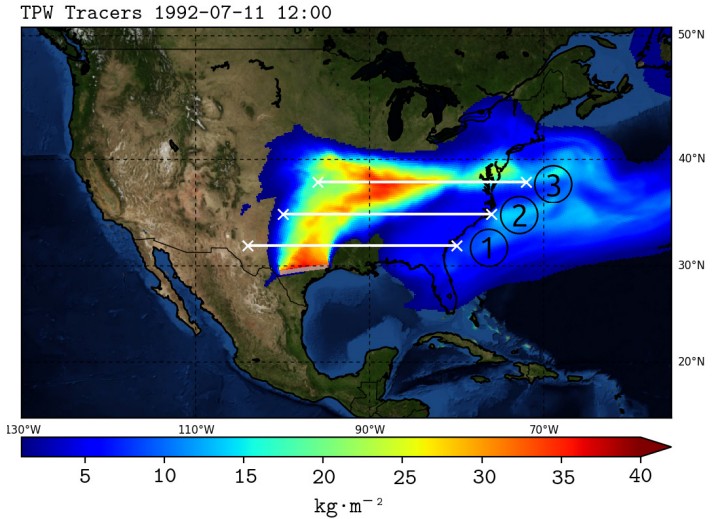

**Figure 6: Tracer total precipitable water (TPW, g kg⁻¹) and positions of the cross sections along the central axis of the GPLLJ shown in Fig. 5, at latitudes 32ºN (1), 35ºN (2) and 38ºN (3) for the main jet event of July 11, 2002.**



**Table 1: Case-studies objectively selected based in the frequency distribution of the LLJ index to carry out WRF-TT simulations. μ**
15  **is the mean of the distribution and σ its standard deviation.**

| Simulation | Gaussian | NLLJ value | Date | Stat. weight |
|:---:|:---:|:---:|:---:|:---:|
| **0** | | 0.00 | 2012-07-12 | 0 |
| **1** | μ - 2σ | 1.49 | 1999-07-19 | 0.0623 |
| **2** | μ - σ | 5.54 | 1983-07-23 | 0.2445 |
| **3** | μ | 10.19 | 1992-07-11 | 0.3864 |
| **4** | μ + σ | 14.54 | 2002-07-28 | 0.2445 |
| **5** | μ + 2σ | 18.89 | 2016-07-14 | 0.0623 |

25





**Appendix A: Statistical weights in the analysis**

**Table A1. Events**

| Simulation | Gaussian Point |
|---|---|
| 1 | $\mu - 2\sigma$ |
| 2 | $\mu - \sigma$ |
| 3 | $\mu$ |
| 4 | $\mu + \sigma$ |
| 5 | $\mu + 2\sigma$ |

**Table A2.** Gaussian fit $y(x) = y_0 + A \cdot exp\left(-\frac{(x-y)^2}{2\sigma^2}\right)$.

| Simulation | Gaussian Point |
|---|---|
| $y_0$ | $0.03 \pm 3.66$ |
| $A$ | $8.63 \pm 4.39$ |
| $\mu$ | $10.19 \pm 0.19$ |
| $\sigma$ | $4.35 \pm 0.30$ |
| $R^2$ | $0.95$ |

$$StatWeight_1 = StatWeight_5 = \frac{\int_{\mu - 2.5\sigma}^{\mu - 1.5\,\sigma} y(x)dx}{\int_{\mu - 2.5\sigma}^{\mu + 2.5\sigma} y(x)dx} = 0.0623$$

$$StatWeight_2 = StatWeight_4 = \frac{\int_{\mu - 1.5\sigma}^{\mu - 0.5\,\sigma} y(x)dx}{\int_{\mu - 2.5\sigma}^{\mu + 2.5\sigma} y(x)dx} = 0.2446$$

$$StatWeight_3 = \frac{\int_{\mu - 1.5\sigma}^{\mu + 0.5\,\sigma} y(x)dx}{\int_{\mu - 2.5\sigma}^{\mu + 2.5\sigma} y(x)dx} = 0.38643$$

