# Peer review of "On the assessment of the moisture transport by the Great Plains lowlevel jet"

_Earth System Dynamics, 2018_

## Referee Comment (RC1) · Anonymous Referee #1 · 26 Nov 2018

This study focuses on the Great Plains low level jet (GPLLJ) and the associated moisture during the month of July. Overall, I believe that this journal represents a good venue for this work, and that this study will be a nice contribution to the literature after some additional analyses and clarifications.

- In the abstract, you mention atmospheric rivers but then there is no further mention of them in the text. Is it really needed to mention them there?

- There have been studies looking at atmospheric rivers and precipitation focusing on specific events across the central United States. The papers by Moore et al. (2012) and Nayak et al. (2016) are likely worth mentioning.

- More broadly, there have been a growing body of work related to moisture transport

over the central United States (e.g., Nakamura et al. 2013; Lavers and Villarini 2015; Steinschneider and Lall 2015, 2016; Nayak and Villarini 2017).

- Pg. 2, line 17: "Higgins et al. (1996)"

- Page 2, line 34: "which modulate a"

- Page 3, lines 17-18: "total amount of total precipitable water" seems a bit redundant. What about "amount of total precipitable water"? The same applies to other places in the text.

- Page 3, line 20: "as follows: in"

- Section 2.1: why is the focus only on July and not on June and August as well? Please clarify.

- Page 3, line 24: "Rife et al. (2010)"

- Page 3, line 29: is there an impact on setting a threshold in terms of these differences? As of now, the only requirement is that the wind speed is higher at midnight compared to midday. What happens if you set a threshold, say 10% higher at midnight compared to midday? Would it be possible to have information related to the distribution of the differences between them?

- Page 3, line 30: "than above it ($\sim$4km)"? I still don't think it is the right wording but it sounds a bit better than what is there.

- Page 3, line 32: "Because of the jet"

- Page 4, line 9: "variation of 5 the wind" is not clear.

- Page 4, line 17: "possible to calculate"

- Page 4, line 30: "followed for 10"

- Page 5, line 12: "above the 75th percentile"

- Page 5, line 16: "is used to quantify the"

- Is it possible to show some results related to the validation of the WRF model with respect to observations?

- Page 6, line 1: the symbol phi for the instant flux of moisture was already used in equation 3.

- Page 6, line 16: the use of the correlation coefficient is not appropriate. Please use the Lilliefors test to test whether the data can be described by a Gaussian distribution. Another option is the Jarque-Bera test.

- Page 6, line 19: "spans 11 days"

- I would remove the equation from the caption of Figure 1.

- Pg. 7, line 5: shouldn't this be Figure 2 instead of Figure 1?

- Page 7, line 27: why not computing the climatology using all the days, rather than based on just a handful?

References:

Lavers, D.A., and G. Villarini, The contribution of atmospheric rivers to precipitation in Europe and the United States, Journal of Hydrology, 522, 382-390, 2015.

Moore, B.J., P.J. Neiman, F.M. Ralph, and F.E. Barthold, Physical processes associated with heavy flooding rainfall in Nashville, Tennessee, and vicinity during 1–2 May 2010: The role of an atmospheric river and mesoscale convective systems, Monthly Weather Review, 140, 358–378, 2012.

Nakamura, J., U. Lall, Y. Kushnir, A.W. Robertson, and R. Seager, Dynamical structure of extreme floods in the U.S. Midwest and the United Kingdom, Journal of Hydrometeorology, 14, 485–504, 2013.

Nayak, M.A., G. Villarini, and A.A. Bradley, Atmospheric rivers and rainfall during

NASA's Iowa Flood Studies (IFloodS) campaign, Journal of Hydrometeorology, 17(1), 257-271, 2016.

Nayak, M.A, and G. Villarini, A long-term perspective of the hydroclimatological impacts of atmospheric rivers over the central United States, Water Resources Research, 53, 1144-1166, 2017.

Steinschneider, S., and U. Lall, Daily precipitation and tropical moisture exports across the Eastern United States: An application of archetypal analysis to identify spatiotemporal structure, Journal of Climate, 28(21), 8585–8602, 2015.

Steinschneider, S., and U. Lall, Spatiotemporal structure of precipitation related to tropical moisture exports over the eastern United States and its relation to climate teleconnections, Journal of Hydrometeorology, 17(3), 897–913, 2016.

---

## Referee Comment (RC2) · Anonymous Referee #2 · 27 Nov 2018

The following is a review of the article titled "On the assessment of the moisture transport by the Great Plains low-level jet" by Algarra et al. The article details a short study that is limited but well-defined and effective in demonstrating the utility of the WRF 3D water vapor tracers scheme as applied to southerly Great Plains LLJs. A Lagrangian FLEXPART model is used to identify the Gulf source region and from the ERA-INT distribution of GPLLJ strength as classified according to Rife et al. (2010), five representative case events are selected for detailed WRF-TT studies. For these five cases plus a control, non-GPLLJ case, the authors quantify the ratio of GPLLJ-sourced moisture/moisture flux across the continental U.S. A few selective latitudinal vertical cross section composites included provide a window into the vertical transport of GPLLJ moisture over time and space. The article is generally well-written, although a clear

paragraph structure is lacking in parts (i.e., several apparent floating sentences) and some misspelling occurs (e.g., norheastern for northeastern). I provide a few general and specific comments below that should be addressed before publication.

General Comments:

1) The FLEXPART analysis, to my understanding, is largely underutilized here. Is it only to justify the location of the wall line? I think the wall line location is fairly intuitive and I don't believe the authors would find much sensitivity to its location (within reasonable limits). As a minimum, I would encourage the authors to include the FLEXPART-derived moisture source regions for each case study in Supplemental Material.

2) The article focuses entirely on July GPLLJs with the logic that southerly GPLLJ frequency is highest for this month. Firstly, have the authors found this to be the case? In my own work, I have found May to be the month of highest frequency. The authors should include a figure or table of the ERA-INT-derived monthly GPLLJ climatology. Secondly, are July GPLLJs representative of the springtime LLJs that are predicted to increase in frequency and intensity (lns 6-10, ph 3)? The authors could have designed their study to be better aligned with their motivations/stated best projections of a future GPLLJ.

Specific Comments:

Abstract: mention of ERA-INT and "southerly" GPLLJ needs to be made

Introduction: the work of Claudia Walters and Julie Winkler on GPLLJ (northerly and southerly) climatologies needs to be referenced here. There are several works from which to choose between 2001-present.

Pg2,ln12 insert "southerly"

Pg2,ln21 specify whether Higgins et al (1997b) analysis was conditioned on GPLLJ occurrence

Pg2,ln23 unclear meaning of "compared with the diurnal one"

Pg2,ln24 unclear if "this work" refers to Higgins or Mo reference

Pg2,ln27 suggest "found regional correlation at a distance between…" or similar

Pg2,lns30-32- one example of a "floating sentence" that needs to be grouped with another paragraph

Pg2ln31 meaning of "local" is unclear. Define local as opposed to non-local in this context.

Pg3,ln4 suggest replacing "common" with "frequent"

Pg3,lns3-10 more floating sentences

Pg3,ln17 word "total" may be deleted

Pg3,ln19 reword "and the at the point"

Pg3,ln27 on a monthly basis, I believe the max GPLLJ frequency is in May

Pg3,ln32 the native resolution of ERA-INT is closer to 0.75deg. How was it spatially interpolated (oversampled) to 0.25deg resolution?

Pg5,ln29 delete "30"

Pg6,ln17 clarify for the reader whether these events were chosen from the NLLJ distribution at 32.75N,99W or for the regional distribution (w/I cyan outline)

Pg6,ln25 should "LLJ" be "NLLJ"?

Pg7,ln22 "northeastern"

Pg8,ln8 clarify that this is done for a specific point (32.75N,99W)

Pg8,ln16 I do not believe it is true that GPLLJ occurs on more than 16/31 nights in July. Please quantify this using ERA-INT.

Pg8,ln20 "northeastern"

Pg8,ln21 synoptic and land preconditioning will impact ratio of GPLLJ TPW (Fig 3).

Pg8,ln26 replace "leaded" with "preceded"

Pg8,ln31 suggest "...North America [using WRF-TT. Additional] simulations should..."

Fig. 1 the cyan color is hard to distinguish in my color print. Clarify whether these "frequency distributions" are derived for the region contained in the cyan outline or for a single point (i.e., 32.75N, 99W).

Figs 2-4. Lat/lon labels required on these figures.

Fig 4. Suggest adding state boundaries

Fig5-6. The order of Fig5 and Fig6 should be switched. Would it also be informative to plot the vertical cross section of relative GPLLJ humidity? E.g., qTR:q; phiTR;phi?

Table 1. Specify ERA-INT-derived as well as the lat/lon location or domain over which the frequency distribution was composed.

---

## Author Comment (AC1) · 23 Dec 2018

We thank the reviewer for all his/her comments which will help to improve the manuscript substantially. We want to thank you for the references provided in the review. Please, find below the response to your comments, questions, and suggestions.

- In the abstract, you mention atmospheric rivers but then there is no further mention of them in the text. Is it really needed to mention them there? - In the abstract we had mentioned the atmospheric rivers (ARs) because together with low-level jets (LLJ), they are the main structures that are associated with a transport of high atmospheric moisture and can trigger episodes of heavy rains or even floods. We agree with the reviewer that this may be misplaced and we will remove the reference in the latest version of the manuscript.

- There have been studies looking at atmospheric rivers and precipitation focusing on specific events across the central United States. The papers by Moore et al. (2012) and Nayak et al. (2016) are likely worth mentioning. – Both references proposed by the reviewer will be added to the introduction.

- More broadly, there have been a growing body of work related to moisture transport over the central United States (e.g., Nakamura et al. 2013; Lavers and Villarini 2015; Steinschneider and Lall 2015, 2016; Nayak and Villarini 2017). – Most of the references proposed by the reviewer will be included in the latest version of the manuscript.

- Pg. 2, line 17: "Higgins et al. (1996)" – The typo has been corrected.

- Page 2, line 34: "which modulate a" – The typo has been corrected.

- Page 3, lines 17-18: "total amount of total precipitable water" seems a bit redundant. What about "amount of total precipitable water"? The same applies to other places in the text. – The text has been updated following the suggestion of the reviewer.

- Page 3, line 20: "as follows: in" – The typo has been corrected.

- Section 2.1: why is the focus only on July and not on June and August as well? Please clarify. - The Great Plains low-level jet (GPLLJ) is a phenomenon mostly related to the warm season. We have focused the study in the month of July because it is the month of the year when we find the highest frequency. Figure attached below shows the monthly distribution of GPLLJ detections.

[Figure]

- Page 3, line 24: "Rife et al. (2010)" – The typo has been corrected in the new version of the manuscript.

- Page 3, line 29: is there an impact on setting a threshold in terms of these differences? As of now, the only requirement is that the wind speed is higher at midnight compared to midday. What happens if you set a threshold, say 10% higher at midnight compared to midday? Would it be possible to have information related to the distribution of the differences between them?

The index is based on the temporal variation of the vertical wind structure. To obtain a day of GPLLJ it is necessary that two conditions are simultaneously met:

1. The wind speed is higher at midnight than at local noon.
2. The wind speed on the surface is higher than at high levels.

We do not believe it is adequate to apply a threshold in the detection of LLJ. The fact of establishing a threshold in one of the conditions would add subjectivity to the methodology used in the study. Nonetheless, if we set the arbitrary threshold of 10% in one of the conditions, the climatology however hardly changes. We have performed the calculation and applying the 10% threshold at midnight we identified 924 days of GPLLJ. However, without applying the threshold we get 931 days of GPLLJ (7 cases of difference). Have or not a day of GPLLJ is mainly due to the fulfilment of both initial conditions. Setting random thresholds only in one condition adds subjectivity to the methodology used in the study.

- Page 3, line 30: "than above it (_4km)"? I still don't think it is the right wording but it sounds a bit better than what is there. – Following the reviewer's suggestion, the sentence now reads as follow: *The local midnight wind speed is higher at the surface (~ 500m) than above it (~ 4km).*

- Page 3, line 32: "Because of the jet".

- Page 4, line 9: "variation of 5 the wind" is not clear.

- Page 4, line 17: "possible to calculate".

- Page 4, line 30: "followed for 10".

- Page 5, line 12: "above the 75th percentile".

- Page 5, line 16: "is used to quantify the".

The typos detected by the reviewer have been corrected in the latest version of the manuscript.

- Is it possible to show some results related to the validation of the WRF model with respect to observations?

We understand the reviewer is asking for some validation or comparison with observed precipitation, which is the most "tricky" variable to solve by all models, including WRF. Figure attached below compares 11-days accumulated precipitation for WRF simulations versus CPC gauge-analysis observations (https://www.esrl.noaa.gov/psd/data/gridded/data.unified.daily.conus.html) throughout the same periods. As the reviewer can note, WRF tends to slightly overestimate the precipitation, but it is in all respects represented quite well, both in amount and field distribution. This figure will be added to the supplementary material.

[Figure]

11-day accumulated precipitation
10  15  20  30  40  50  75  100  125  150  200
mm

- Page 6, line 1: the symbol phi for the instant flux of moisture was already used in equation 3. – "phi" has been replaced by "sigma" in the latest version of the manuscript.

- Page 6, line 16: the use of the correlation coefficient is not appropriate. Please use the Lilliefors test to test whether the data can be described by a Gaussian distribution.

Another option is the Jarque-Bera test.

We agree with the referee on the fact that the correlation coefficient is not an appropriate indicator of the normality of the distribution. Thus, we have applied the Jarque-Bera test which provided a p-value equal to 0.0055. We understand that this p-value is low enough for considering the LLJ distribution as Gaussian.

Accordingly, "A clear peak around 11 m s$^{-1}$ is observed together with a Gaussian behaviour ($R^2$ = 0.95, red line)" has been replaced by:

"A clear peak around 11 11 m s$^{-1}$ is observed together with a Gaussian behaviour (Jarque-Bera test p-value=0.0055, which provides a confidence level close to 99.5 %, red line)"

- Page 6, line 19: "spans 11 days".

- I would remove the equation from the caption of Figure 1.

- Pg. 7, line 5: shouldn't this be Figure 2 instead of Figure 1?

The manuscript has been updated following the Reviewer's suggestions.

- Page 7, line 27: why not computing the climatology using all the days, rather than based on just a handful? - The calculation of the climatology using every day of GPLLJ would increase the computational cost in an excessive way. The methodology used in this work, especially the Eulerian model WRF, has a high computational cost. Besides, this multiply the number of simulations making this work unaffordable. In addition, the aim to this study is to quantify the average transport of moisture in a general perspective of the GPLLJ's behaviour. Fig. 4 shows an approximation to this result. This figure is calculated based on the statistical weight of each simulation of the 5 GPLLJ events analysed. Thus, this figure can be understood as a climatological approach to the moisture transport associated with the GPLLJ.

References:

1. Lavers, D.A., and G. Villarini, The contribution of atmospheric rivers to precipitation in Europe and the United States, Journal of Hydrology, 522, 382-390, 2015.
2. Moore, B.J., P.J. Neiman, F.M. Ralph, and F.E. Barthold, Physical processes associated with heavy flooding rainfall in Nashville, Tennessee, and vicinity during 1–2 May 2010:

The role of an atmospheric river and mesoscale convective systems, Monthly Weather Review, 140, 358–378, 2012.

3. Nakamura, J., U. Lall, Y. Kushnir, A.W. Robertson, and R. Seager, Dynamical structure of extreme floods in the U.S. Midwest and the United Kingdom, Journal of Hydrometeorology, 14, 485–504, 2013.

4. Nayak, M.A., G. Villarini, and A.A. Bradley, Atmospheric rivers and rainfall during NASA's Iowa Flood Studies (IFloodS) campaign, Journal of Hydrometeorology, 17(1), 257-271, 2016.

5. Nayak, M.A, and G. Villarini, A long-term perspective of the hydroclimatological impacts of atmospheric rivers over the central United States, Water Resources Research, 53, 1144-1166, 2017.

6. Steinschneider, S., and U. Lall, Daily precipitation and tropical moisture exports across the Eastern United States: An application of archetypal analysis to identify spatiotemporal structure, Journal of Climate, 28(21), 8585–8602, 2015.

7. Steinschneider, S., and U. Lall, Spatiotemporal structure of precipitation related to tropical moisture exports over the eastern United States and its relation to climate teleconnections, Journal of Hydrometeorology, 17(3), 897–913, 2016.

---

## Author Comment (AC2) · 23 Dec 2018

We thank the reviewer for all his/her questions and suggestions, which will help to improve the manuscript substantially. Please, find attached below the one-to-one reply to the comments.

**Comment 1:**

1) The FLEXPART analysis, to my understanding, is largely underutilized here. Is it only to justify the location of the wall line? I think the wall line location is fairly intuitive and I don't believe the authors would find much sensitivity to its location (within reasonable limits). As a minimum, I would encourage the authors to include the FLEXPART-derived moisture source regions for each case study in Supplemental Material.

We appreciate the comment and the figure of moisture sources for each case study will be added to the supplementary material. Answering the question, the lagrangian Flexpart model was used to objectively justify the position of the wall. Fig. RC2.1 (in this document) shows the 75th percentile the sources of moisture (E - P > 0) for the 6 cases of study. The purple line shows the position of the wall of moisture labelling. The vectors refer to the wind for each case study and are plotted at surface level (aprox. 500m). As it is observed, moisture enters the Great Plains in a channelized way through the wall from the Gulf of Mexico. In the Fig. RC2.1a (no-LLJ day) the vectors do not show the flow of the South. Therefore, the transport of moisture from the Gulf of Mexico will be weakened as is the case with simulation 0. In the remaining simulations we can observe a structure of LLJ with a maximum of moisture transport intensified to the south of the Great Plains responsible for advection of atmospheric moisture from the Gulf of Mexico.

[Figure]

**Figure RC2.1.** Moisture sources for the 6 study cases analysed. The vectors show the wind direction and intensity for each case study at surface level (aprox 500m). The purple line shows the position of the moisture labelling wall used in the WRF simulation.

2) The article focuses entirely on July GPLLJs with the logic that southerly GPLLJ frequency is highest for this month. Firstly, have the authors found this to be the case? In my own work, I have found May to be the month of highest frequency. The authors should include a figure or table of the ERA-INT-derived monthly GPLLJ climatology. Secondly, are July GPLLJs representative of the springtime LLJs that are predicted to increase in frequency and intensity (lns 6-10, ph 3)? The authors could have designed their study to be better aligned with their motivations/stated best projections of a future GPLLJ.

We use the data from ERA-Interim and temporal period of 37 years (from 1980 to 2016). Examining the monthly variability, we find that July is the highest frequency month of GPLLJ. Understood the frequency as the largest number of days of GPLLJ (Fig. RC2.2, in this document). For the 37 years, we detected that the frequency increases during the months of May to October, with more than half of the days of GPLLJ. Nevertheless, we detected a peak in the month of July, with a frequency greater than 80%. Other authors, such as Rife et al. 2010 reported a frequency of GPLLJ of 78% also for the month of July. Despite this author used another dataset and a different time period than the one we use in our work.

[Figure]

**Figure RC2.2** Monthly variability of the percentage of GPLLJ days.

Regarding the second question, the current literature foresees an increase both in the intensity and frequency of GPLLJ in spring. Nevertheless, the main goal of this work is to provide a first quantitative approach to moisture transported by the GPLLJ. Despite the numerous works related with GPLLJ, no work has objectively quantified the moisture associated with this structure. In the introduction we want to highlight the importance of studying these structures related to moisture transport. Changes in frequency and intensity will affect the sink regions and extreme events (droughts and floods). Therefore, it is of great interest to objectively estimate the moisture transported by the GPLLJ. However, in this paper we want to offer a first approximation in terms of average moisture transport. For this reason, we have objectively chosen objectively 5 cases of study based on the Gaussian distribution with the aim to quantify the average transport of moisture in a general perspective of the GPLLJ's behaviour.

**Specific Comments:**

Abstract: mention of ERA-INT and "southerly" GPLLJ needs to be made. - The word *southerly* was added and we mention the ERA-Interim reanalysis data used in the work.

Introduction: the work of Claudia Walters and Julie Winkler on GPLLJ (northerly and southerly) climatologies needs to be referenced here. There are several works from which to choose between 2001-present. - The work of Walters and Winkler, 2008 and other recent works have been added to the introduction.

Pg2,ln12 insert "southerly". – The word "southerly" has been included in the latest version of the manuscript.

Pg2,ln21 specify whether Higgins et al (1997b) analysis was conditioned on GPLLJ occurrence -

Pg2,ln23 unclear meaning of "compared with the diurnal one"

Pg2,ln24 unclear if "this work" refers to Higgins or Mo reference

We consider the last three comments and the paragraph now read as follows:

*Nevertheless, a large number of studies have documented the relationship between the major moisture transport and the GPLLJ. Higgins et al. (1996) studied the moisture budget over the central US in May employing NASA/DAO and NCEP/NCAR datasets, together with station observations, to evaluate the limitations of these products. Although both reanalyses overestimate daily mean precipitation rates, they accurately capture the basic temporal and structural characteristics of the GPLLJ. From the data, these authors calculated an increase in atmospheric moisture transport from the Gulf of Mexico during night-time of more than 50%. In a later work, Higgins et al. (1997) observed a well-defined nocturnal maximum of precipitation over the Great Plains in spring and summer by analysing station data. Particularly and linked to LLJ events this research found in this region an excess of 25% in nocturnal rainfall during summer when compared with the diurnal precipitation, associated with a rainfall decrease over the Gulf of Mexico. Additionally, Higgins et al. (1997) reported significant differences in precipitation pattern in coincidence (or not) with LLJ events. When a LLJ event occurs, the observations show an enhanced precipitation over the north-central United States and the Great Plains region, together with a decrease along the Gulf of Mexico and the western Atlantic. On the other hand, Mo and Juang (2003) found regional correlation at a distance between evaporation and precipitation, reflected in evaporation anomalies over the Great Plains along the trajectory of the GPLLJ, which are associated with downstream precipitation anomalies.*

Pg2,ln27 suggest "found regional correlation at a distance between…" or similar – The text has been updated following the reviewer's suggestion.

Pg2,lns30-32- one example of a "floating sentence" that needs to be grouped with another paragraph – Following the reviewer's suggestion, the paragraph now reads as follows:

*Otherwise, extreme rainfall events in the central US are related to an increase in moisture convergence downwind of the GPLLJ (Mo et al., 1997). A decisive factor that triggers heavy rains and floods is the presence of moisture advected by the GPLLJ from Gulf of Mexico and the Caribbean Sea. Moore et al. (2012) reported the physical processes related to the floods in May, 2010. A persistent southerly low-level jet associated with an atmospheric river (AR) enhanced the transport of moisture from the Gulf of Mexico into the heavy rainfall region. Thus, important socioeconomic impacts follow enhanced GPLLJ events, which modulate a large percentage of the local extreme precipitation events and flooding in warmer months (Mo et al., 1995, 1997; Beljaars et al., 1996; Trenberth and Guillemot, 1996; Arritt et al., 1997; Nakamura et al., 2013;*

*Nayak et al., 2016). All these results are consistent with the large-scale atmospheric moisture transport and support the marked influence of the GPLLJ over the central-eastern US, which has been shown to trigger more than 60% of the spring local precipitation there (Wang and Chen, 2009).*

Pg2ln31 meaning of "local" is unclear. Define local as opposed to non-local in this context. - The sentence has been rewritten as follow: *All these results are consistent with the large-scale atmospheric moisture transport and support the marked influence of the GPLLJ over the central-eastern US, which has been shown to trigger more than 60% of the spring precipitation over the Great Plains region (Wang and Chen, 2009).*

Pg3,ln4 suggest replacing "common" with "frequent" – "common" has been replaced by "frequent".

Pg3,lns3-10 more floating sentences – The cited paragraph now reads as follows:

*During the last decades, the GPLLJ has experienced a strengthening, accompanied by a northward migration causing a displacement of rainfall in the same direction. As a result, more frequent droughts have been observed in the southern Great Plains (Barandiaran et al., 2013). Besides, the increase in the number and intensity of GPLLJ events is also forecasted for future projections, which reveal an intensification of the GPLLJ during the spring season associated with global warming (Cook et al., 2008; Tang et al., 2017). As a result, increasing amounts of moisture transport and rainfall are expected, particularly from April to July, over the central US (Harding and Snyder, 2014). The same projections forecast a slight weakening of the GPLLJ from August to December, which could translate into increasing drought conditions.*

Pg3,ln17 word "total" may be deleted – The word "total" has been removed from the manuscript.

Pg3,ln19 reword "and the at the point" - The sentence has been reworded as *a 37-year climatology was previously calculated at the point of maximum jet intensity...*

Pg3,ln27 on a monthly basis, I believe the max GPLLJ frequency is in May - We have calculated the monthly frequency for the 1980-2016 study period. As we show in figure RC2.2, we obtain the maximum frequency in the month of July.

Pg3,ln32 the native resolution of ERA-INT is closer to 0.75deg. How was it spatially interpolated (oversampled) to 0.25deg resolution? - The Era-Int data were downloaded directly from the ERA Interim web at a resolution of 0.25°. Although the original resolution is 0.75°, it was interim allowing the download to several horizontal resolutions. In our study, we obtained by downloading them at a resolution of 0.25. More information about the interpolation can be found in the following link: https://confluence.ecmwf.int/display/CKB/ERA-Interim%3A+What+is+the+spatial+reference

Pg5,ln29 delete "30" – The typo has been deleted.

Pg6,ln17 clarify for the reader whether these events were chosen from the NLLJ distribution at 32.75N,99W or for the regional distribution (w/I cyan outline) – The sentence: *The five case-studies were selected based on the Gaussian adjustment applied to the study.* has been included in the new version of the manuscript.

Pg6,ln25 should "LLJ" be "NLLJ"? – This typo has been corrected.

Pg7,ln22 "northeastern" – The typo has been corrected.

Pg8,ln8 clarify that this is done for a specific point (32.75N,99W) - The core (32.75°N,99°W) is used only to calculate the 37-yr climatology. We add the following sentence to the text to clarify

it: *The target region used in FLEXPART was defined based on the 75th percentile of the index value.*

Pg8,ln16 I do not believe it is true that GPLLJ occurs on more than 16/31 nights in July. Please quantify this using ERA-INT. - We use the ERA-Interim data for the detection of GPLLJ. As we discussed, we obtained a maximum frequency in July. Please, we refer you to the previous comment.

Pg8,ln20 "northeastern" – The typo has been corrected.

Pg8,ln21 synoptic and land preconditioning will impact ratio of GPLLJ TPW (Fig 3). – This clarification has been included in the text.

Pg8,ln26 replace "leaded" with "preceded" – "leaded" has been replaced by "preceded".

Pg8,ln31 suggest ": : :North America [using WRF-TT. Additional] simulations should…" – "WRF-TT" has been removed from the manuscript.

Fig. 1 the cyan color is hard to distinguish in my color print. Clarify whether these "frequency

[Figure]

a)                                                          b)

distributions" are derived for the region contained in the cyan outline or for a single point (i.e., 32.75N, 99W). - The cyan color is replaced by the magenta. We think it highlights a little better. The frequency of distribution was calculated at the point of maximum intensity. We have been modified the caption of the figure 2 to clarify it.

**Figure 1.** (a) Mean NLLJ index (shaded) and 500 m winds (arrows, in m s$^{-1}$) at local midnight in July (boreal summer) for 1980-2016, calculated from ERA-Interim reanalysis. The black cross at 32.75ºN, 99ºW shows the point of maximum NLLJ in the climatology. The cyan contour line surrounds the region containing points above the 75th percentile. (b) Frequency distribution of the GPLLJ for the months of July from 1980 to 2016 (blue bars). The red curve corresponds to the Gaussian fit (see table A2). Noted: The frequency distribution is calculated at the point of maximum intensity of NLLJ (at 32.75ºN, 99ºW, black cross in fig. 1a).

Figs 2-4. Lat/lon labels required on these figures.

Fig 4. Suggest adding state boundaries.

Lat and lon and the states boundaries have been included in the lasted version of the manuscript. An example figure is shown:

[Figure]

Fig5-6. The order of Fig5 and Fig6 should be switched. Would it also be informative to plot the vertical cross section of relative GPLLJ humidity? E.g., qTR:q; phiTR;phi? – Figure 5 and 6 will be swapped. Additionally, the vertical cross sections will be included and commented in the manuscript.

Table 1. Specify ERA-INT-derived as well as the lat/lon location or domain over which the frequency distribution was composed. - The title of the table has been rewritten as follows:

*Table 1: Case-studies objectively selected based in the frequency distribution of the LLJ index to carry out WRF-TT simulations. μ is the mean of the distribution and σ its standard deviation. Noted: The frequency distribution is calculated at the point of maximum intensity of NLLJ at 32.75ºN, 99ºW (black cross in fig. 1a) using the ERA-Interim reanalysis dataset.*